# Rheumatoid Arthritis-Associated Mechanisms of *Porphyromonas gingivalis* and *Aggregatibacter actinomycetemcomitans*

**DOI:** 10.3390/jcm8091309

**Published:** 2019-08-26

**Authors:** Eduardo Gómez-Bañuelos, Amarshi Mukherjee, Erika Darrah, Felipe Andrade

**Affiliations:** Division of Rheumatology, The Johns Hopkins University School of Medicine, Baltimore, MD 21224, USA

**Keywords:** Rheumatoid arthritis, *Porphyromonas gingivalis*, *Aggregatibacter actinomycetemcomitans*, periodontitis, periodontal disease, citrullination, peptidylarginine deiminase, ACPA, anti-CCP

## Abstract

Rheumatoid arthritis (RA) is an autoimmune disease of unknown etiology characterized by immune-mediated damage of synovial joints and antibodies to citrullinated antigens. Periodontal disease, a bacterial-induced inflammatory disease of the periodontium, is commonly observed in RA and has implicated periodontal pathogens as potential triggers of the disease. In particular, *Porphyromonas gingivalis* and *Aggregatibacter actinomycetemcomitans* have gained interest as microbial candidates involved in RA pathogenesis by inducing the production of citrullinated antigens. Here, we will discuss the clinical and mechanistic evidence surrounding the role of these periodontal bacteria in RA pathogenesis, which highlights a key area for the treatment and preventive interventions in RA.

## 1. Introduction

Rheumatoid arthritis (RA) is an autoimmune disease of unknown etiology characterized by synovial inflammation, joint destruction, and high titer autoantibodies [1]. The disease affects 0.5–1% of the adult population worldwide and is associated with increased mortality rates compared with the general population [2]. A microbial origin in RA has been hypothesized for more than a century [3,4,5,6], yet, different to chronic diseases that have an infectious origin, a causal agent for RA has not been identified. Instead, numerous studies support the idea that the etiology of RA is multifactorial with a complex interplay between genetic and environmental factors [7]. Multiple risk factors have been associated with RA, including specific HLA (human leukocyte antigen) alleles, female sex, smoking, obesity, infections, and menopause, among others [8]. Nevertheless, these factors are also common in the general population, which makes it difficult to understand why only few individuals develop RA and not others with similar risk. Even in monozygotic twins, the disease concordance is rather low, varying between 8.8 and 21% [9,10,11,12]. Although stochastic events may explain the discordance in disease development among individuals at risk, the current model of RA relies on the contribution of an environmental factor, most likely a microbial agent, which may be responsible for triggering the disease in susceptible individuals. 

Assuming that a microorganism is involved in RA pathogenesis, at least two hypotheses may explain why this causal agent has not yet been found. First, RA may be triggered by a rare pathogen that is exclusive to patients with RA, which has not yet been discovered because of the lack of proper technologies. Second, different to other chronic infectious diseases that are linked to a single pathogen, RA may result from the individual or interacting effects of different microbial agents likely unified by triggering common arthritogenic pathways. Moreover, like other risk factors in RA, microbial species with arthritogenic potential may also be common in the general population, but only drive RA in the perfect setting of other predisposing elements.

While it cannot be excluded that a single unknown pathogen may cause RA, current studies suggest that dysbiotic microbiomes may play a role in the pathogenesis of the disease [13,14,15]. In this regard, several bacterial candidates have been mechanistically linked to RA either by creating a pro-inflammatory environment or by inducing the production of autoantibodies [13,14,15]. Here, we will discuss the evidence surrounding the potential role of two of these microbial agents, *Porphyromonas gingivalis* (*P. gingivalis*) and *Aggregatibacter actinomycetemcomitans* (*Aa*), in the mechanistic model of RA. Importantly, these pathogens are not exclusive to RA. Indeed, they are causal agents of periodontitis, a common disease in the adult population initially associated with RA in the late 1800s [16].

## 2. The Focal Infection Theory, Oral Sepsis, and RA

In order to discuss the potential role of periodontal pathogens in the etiology of RA, it is important to first review the historical background leading to the link between the mouth and RA. In contrast to other arthritides known for several centuries, the first description of RA (initially termed “primary asthenic gout”) was not made until 1800 by Augustin Jacob Landré-Beauvais [17]. Later, Alfred Garrod in 1859 proposed the name “rheumatoid arthritis” for diseases previously known as chronic rheumatic arthritis and rheumatic gout, and provided a detailed clinical description to distinguish RA from gout [18]. Once RA was established as an independent entity, numerous investigators tried to identify the etiology of this disease. Due to the inflammatory nature of RA, it was thought that the etiology was infectious, leading to the search for microbial organisms in the joints of patients with RA.

Between 1887 and the early 1900s, several investigators claimed to identify different bacteria in RA synovial fluid and tissue [3,4,5,6]. Although the results were inconsistent and not reproducible [19,20,21,22], these initial studies reinforced the idea that RA had an infectious origin, either due to bacteria within the joint itself or to a toxin produced by microorganisms in some other part of the body [23]. During the late 1800s, it was widely accepted that an infection in one part of the body may have effects on a different anatomical site (e.g., the syphilitic chancre is followed by systemic infection [24]). Thus, this notion fueled speculation that similar to other chronic diseases [25], RA was the result of microbial dissemination from distant sites of chronic focus of infection, including nasal sinuses, gut, lungs, genitourinary tract and the mouth [26,27,28,29,30]. This idea, known as the focal infection theory, defined the cause and treatment of a broad range of diseases, including RA, for several decades in the early 1900s [25,31,32].

In 1891, Willoughy D. Miller first conjectured that oral pathogens play a significant role in the production of local and systemic diseases [33,34]. This notion was further promoted by William Hunter in 1900, who was responsible for highlighting that the mouth was a major focus of infection driving diseases in other organs of the body. In particular, he reached the incorrect conclusion that pernicious anemia was the result of infective gastritis caused by oral sepsis [35,36,37,38]. Following these observations, oral sepsis (including caries, gingivitis, stomatitis, periodontitis, and tonsillitis) was considered a major focus of infection causing systemic diseases in which the etiology was unknown, such as RA [25,30,39,40]. This notion was further supported by anecdotal reports of patients with “rheumatism” cured after the removal of carious teeth [41], leading to the use of tonsillectomies and tooth extractions as the standard of treatment [29,31]. In the 1930s, this theory was refuted by the demonstration that neither tooth extraction nor tonsillectomy provided clinical benefit to patients with RA [42,43,44].

## 3. Periodontitis and RA

Among the different causes of oral sepsis that were linked to RA, an association with “pyorrhea alveolaris” was noticed in 1895 [16]. Pyorrhea alveolaris, also known as Riggs’ disease and currently termed periodontal disease (PD) or periodontitis, was described by John W. Riggs in 1875 [45]. Periodontitis is a bacterial-induced chronic inflammatory disease affecting the tissues that support the teeth, including the alveolar bone. The disease results from dysbiosis of the oral microbiota and is associated with destruction of periodontal tissue, potentially leading to tooth loss [46]. Since the implantation site of the teeth is also an articulation, it was initially thought that the destruction of the periodontal membrane and alveolar bone in patients with RA and pyorrhea alveolaris was part of same RA-induced damage affecting the dentoalveolar joint [16]. Nevertheless, the establishment of the focal infection theory tilted the paradigm to underscore periodontitis as a cause (not as a consequence) of RA in the early 1900s [26,27,28].

However, after the downfall of the focal infection theory as the cause of RA in the 1930s, the study of periodontitis in the pathogenesis of RA was left behind. Later, at least two major advances renewed interest in periodontitis as a component of the mechanistic model of RA. The first occurred between the 1960s–1990s, when significant advances were made in the immunopathogenesis of periodontitis, which suggested important mechanistic similarities with autoimmune diseases [47], in particular RA [48]. These include common mechanisms of immune-mediated tissue damage, patterns of pro-inflammatory cytokines (e.g., interleukin (IL)-1, IL-6 and tumor necrosis factor α), immune complex deposition and complement activation [48,49,50]. Rapidly progressive periodontitis is also associated with *HLA-DRB1* alleles linked to RA [51,52]. Moreover, B cells and plasma cells predominate in the affected sites in chronic periodontitis [53,54,55], autoantibodies such as rheumatoid factor (RF) and anti-collagen antibodies are found in the periodontal lesion [56,57,58,59], and RF can be detected in dental periapical lesions from patients with RA [60]. The second advance was in 1999, when it was discovered that an important periodontal pathogen (*P. gingivalis*) secretes a peptidylarginine deiminase (PAD)-like enzyme [61], which was incorrectly thought to be equivalent to human PADs at the time [62]. As PAD enzymes are responsible for generating citrullinated proteins, major targets of autoantibodies in RA [63], it was hypothesized that periodontitis drives RA via the production of citrullinated antigens by *P. gingivalis* [62]. Together, these findings have provided the basis for the renewed theory that periodontitis and RA may be mechanistically related and potentially linked by a common etiologic factor.

In the last 10 years, numerous epidemiological studies, extensively reviewed elsewhere [64,65,66], have reported a positive association of RA with PD when compared to healthy (non-RA) controls. Overall, a recent meta-analysis found that patients with RA had a 13% greater risk of periodontitis compared to healthy controls, ranging from 4 to 23% (RR: 1.13; 95% CI: 1.04, 1.23; *p* = 0.006) [65]. In addition, a case-control study from the Medical Biobank of Northern Sweden found that periodontitis, characterized as marginal jawbone loss, precedes the clinical onset of RA [67], supporting a potential role for PD in RA pathogenesis. Not every study, however, has confirmed this association either by comparing RA with healthy controls [68,69] or with patients with osteoarthritis (OA) [65]. Although these studies have methodological differences that may explain their discrepancies, a causal relation between RA and periodontitis may be difficult to sustain based purely on association studies. 

A major caveat in the epidemiological association between RA and periodontitis is that PD is likely the most frequent chronic infectious disease in humans worldwide. The overall rate of PD in the adult US population is 47%, with 38% over age 30 and 64% over age 65 having either severe or moderate periodontitis [70]. Moreover, severe forms of periodontitis affect 11.2% of the global adult population [71]. Considering that almost half of the adult population has some form of PD, it may be hard to demonstrate a causal relationship with RA, since its prevalence is only 0.5–1% of the adult population [2]. Indeed, the relative risk of periodontitis in patients with RA is only 1.13 when compared to healthy controls, and of 1.10 compared to OA [65]. Despite these potential shortcomings, additional studies have been centered on addressing whether periodontitis, and in particular periodontal pathogens, may have a mechanistic role in RA through the production of citrullinated antigens.

## 4. Citrullination and RA

The discovery that the majority of patients with RA have antibodies to citrullinated proteins (known as ACPAs) [63,72,73] marked an important advance in understanding potential pathogenic mechanisms in RA [1]. Citrullination is an enzymatic process mediated by the peptidylarginine deiminases (PADs) in which arginine residues are deiminated to generate citrulline residues [74]. Five PADs have been identified in humans (PAD1–4 and 6) [1], but only PAD1–4 have citrullinating activity [75]. PAD2 and PAD4 have gained prominence as potential candidates that drive citrullination of self-antigens in RA due to their increased expression in rheumatoid synovial tissue and fluid [76,77,78]. PADs are calcium dependent enzymes. Four, five, and six calcium-binding sites were identified in the structure of PAD1, PAD4, and PAD2, respectively, with calcium binding inducing conformational changes required to generate the active site cleft [79,80,81]. PADs are highly specific for peptidylarginine residues, requiring at least one additional amino acid residue N-terminal to the site of modification [74,82]. Thus, these enzymes can only citrullinate arginine residues within polypeptide chains but not at their termini (i.e., they are endodeiminases). Different from arginine deiminases (ADI), which catalyze the deimination of free L-arginine, PADs cannot generate citrulline from free L-arginine [74]. PADs 2, 3, and 4 form homodimers, whereas PAD1 is monomeric in solution [79,80,81]. Each PAD monomer contains a C-terminal catalytic domain and an N-terminal domain involved in substrate binding and protein–protein interactions [79,80,81]. The PADs are highly conserved and share 50%–55% sequence identity [79], but exhibit distinct substrate preferences and tissue expression [83,84].

Citrullination is a normal process across multiple tissues in humans [85]. More than 200 proteins are citrullinated in different healthy human tissues, with the highest levels found in the brain and lungs [85]. Together, this set of proteins is referred to as the citrullinome. Large amounts of citrullinated proteins are found in RA synovial fluid, including more than 100 proteins that are normally citrullinated among different normal tissues [85,86,87,88,89,90,91]. This unique pattern of citrullination that includes proteins spanning the range of molecular weights is termed hypercitrullination [87]. Similar to the RA joint, it is noteworthy that periodontitis is also characterized by the accumulation of large amounts of citrullinated proteins with similar patterns of hypercitrullination found in RA [92,93], supporting the notion that PD is a potential source of citrullinated autoantigens. Understanding which components in periodontitis are responsible for driving hypercitrullination may, therefore, provide important mechanistic insights into RA pathogenesis.

## 5. *P. gingivalis* in RA Pathogenesis

*P. gingivalis* was linked to periodontal disease in the early 1960s, initially named *Bacteroides melaninogenicus* (*B. melaninogenicus*) [94], which was later found to include two species, *B. melaninogenicus* (later *Prevotela melaninogenica*) and *B. asaccharolyticus* [95]. The name *B. gingivalis* was later proposed to distinguish oral from non-oral *B. asaccharolyticus* [96,97], and in the late 1980s, *B. gingivalis* was further reclassified in a new genus, the *Porphyromonas* [98]. *P. gingivalis* has been implicated in the pathogenesis of periodontitis by subverting host immune defenses, leading to overgrowth of oral commensal bacteria, which causes inflammatory tissue destruction [99,100].

Different approaches have been used to address a potential association between *P. gingivalis* and RA, many of which have shown inconsistent or inconclusive results. These include: (1) bacterial detection in gingival tissue, subgingival plaque and/or gingival crevicular fluid (GCF) either by staining using anti-*P. gingivalis* antibodies, bacterial culture, and/or DNA amplification (PCR) [101,102,103,104,105,106,107]; (2) metagenomic sequencing in saliva and subgingival plaque, which surveys complex microbial communities [103,106,108,109,110,111,112]; and (3) measuring antibodies to *P. gingivalis* in serum [101,113,114,115,116,117,118,119,120,121,122,123,124,125], which in contrast to the other assays that detect the existing bacteria in the mouth, it is an indirect test that determines past or present exposure to *P. gingivalis*.

Due to its convenience, the detection of antibodies to *P. gingivalis* (mainly IgG) has been the most widely used assay to study the relationship between this pathogen and RA. However, it is important to highlight that there is not a standard assay to measure such antibodies. All studies used in house ELISAs with different types of *P. gingivalis* antigens. These include bacterial cells either as intact bacteria (not fixed or fixed with formalin) or bacterial extracts generated by sonication [101,113,114,115,116,117,118,119], purified recombinant proteins (e.g., bacterial PAD, arginine gingipain (RgpB), or hemin binding protein 35 (HBP35)] [120,121,122,123,124,126], HtpG peptides (HSP90 homologue) [111], *P. gingivalis*-specific lipopolysaccharide or outer membrane antigens [125,127]. In regard to the bacterial strains, studies have included either laboratory strains or clinical isolates, which may have some antigenic differences [128]. Considering conflicting conclusions from these studies [101,113,114,115,116,117,118,119,120,121,122,123,124,125] and others not cited in this review, an association between anti-*P. gingivalis* antibodies and RA is difficult to sustain. However, based on a selected number of publications, two meta-analysis have reported a significant association between higher antibody titers to *P. gingivalis* and RA when compared to healthy controls with unknown PD status [129], and to healthy controls with and without PD [130]. Anti-*P. gingivalis* antibody levels also positively correlated with ACPAs in one study [129]. Nevertheless, a significant association was not found between antibodies to *P. gingivalis* and RA when compared to non-RA controls (e.g., population-based case-controls) [129]. This finding may be explained by the presence of individuals with chronic diseases and high prevalence of severe PD in the non-RA population, supporting the idea that the relative abundance of *P. gingivalis* is associated with PD severity regardless of RA [111]. Moreover, one meta-analysis showed that only antibodies to whole *P. gingivalis*, but not anti-RgpB antibodies, were positively associated with RA [129], highlighting that the anti-*P. gingivalis* antibody assays are not comparable. Interestingly, one study reported that anti-RgpB antibody concentrations increase before the onset of RA symptoms [123], suggesting a temporal relationship between *P. gingivalis* infection and the clinical onset of RA.

Unlike the detection of antibodies to *P. gingivalis,* which has shown some positive association with RA, a number of studies detecting *P. gingivalis* by PCR and metagenomic sequencing in saliva, subgingival plaque and/or GCF failed to confirm such an association [103,104,105,106,107,108,109,110,111]. In one of these studies, however, *P. gingivalis* was detected more frequently in a small subgroup of patients with recently diagnosed, never-treated RA, in comparison with patients with established RA or healthy controls; although this finding could be explained by a higher prevalence of severe periodontitis in those patients [111]. More recently, an increase in the relative abundance of *P. gingivalis* was found at healthy periodontal sites in at risk anti-CCP antibody positive individuals compared with anti-CCP negative healthy controls, though the significance of this finding is unclear [112].

From these studies, it is difficult to reach a definitive conclusion as to the relationship between RA and *P. ginigvalis*. A complicating factor is that every assay has different implications with regard to the status of *P. gingivalis* in the patient (e.g., carrier, active infection, or past and present exposure), and their significance is likely influenced by the source of antigen (in the case of antibodies), periodontal sample collection, and the control group used for comparison (e.g., OA, non-RA, healthy controls with or without PD). Overall, it is likely that positive associations with *P. gingivalis* may reflect PD severity rather than RA status, which importantly, neither supports nor excludes a pathogenic role for *P. gingivalis* in RA.

### 5.1. P. gingivalis in Experimental Models of Arthritis

In addition to epidemiological data, experimental studies using different animal models of immune-mediated arthritis (e.g., collagen-induced arthritis (CIA), methylated bovine serum albumin-induced arthritis, and the SKG mouse model) and different routes of bacterial inoculation (including oral, intraperitoneal and subcutaneous chambers) reached the conclusion that *P. gingivalis* can increase the incidence and/or exacerbate the disease [131,132,133,134,135,136,137,138,139,140,141]. 

However, many of these studies have important caveats. Several studies have reported that DBA/1 mice (a susceptible strain to induce CIA) are resistant to oral colonization by *P. gingivalis* and therefore, have recommended the use of non-oral routes of inoculation or different strains of mice to study the effect of *P. gingivalis* in experimental arthritis [131,136,142]. These findings contrast with others that have successfully induced PD with *P. gingivalis* in DBA/1 mice [132,134,135]. Among DBA/1 mice that developed periodontitis, however, there are also differences observed, with some studies showing that PD exacerbates CIA [132,134], but not others [135]. Using the SKG model of spontaneous arthritis, there are also discrepancies in the induction of PD by *P. gingivalis*, and whether this process aggravates arthritis [141,142]. Although these conflicting data may be explained by differences in mouse strains or environmental conditions that may affect the immune system in mice, the data certainly highlights the lack of reproducible mouse models to study the effect of periodontitis and *P. gingivalis* in autoimmune arthritis.

Using models of arthritis in rats, periodontitis induced by *P. gingivalis* had no effect in the development or severity of pristane-induced arthritis in Dark Agouti rats [143]. More recently, however, one study showed that *P. gingivalis*-associated periodontitis was sufficient to induce erosive arthritis in Lewis rats, providing the first direct evidence that *P. gingivalis* may have the capacity to induce arthritis [144]. 

Several mechanisms have been proposed by which *P. gingivalis* may promote autoimmune arthritis in humans and in experimental animals. These include the induction of a systemic Th17 cell response [132,133,134], induction of autoantibodies [131,138,143,144], cross-reactivity between bacterial and host antigens [145], by promoting C5a generation [141], through direct dissemination of *P. gingivalis* into the joints [136], and by swallowing the bacteria, which may alter the gut microbiota and gut immune system [137]. Nevertheless, the production of citrullinated autoantigens by a PAD enzyme released from *P. gingivalis* is considered the most important mechanistic evidence to support a role of *P. gingivalis* in RA pathogenesis [14,146]. 

### 5.2. P. gingivalis PAD (PPAD)

The existence of an arginine deiminase-like enzyme in *P. gingivalis* was suggested in the early 1990’s [147], and the enzyme was purified and characterized in 1999 [61]. PPAD is not evolutionarily related to mammalian PADs. Structurally, it is a close relative of agmatine deiminases, which are found across bacteria [148]. Different to mammalian PADs, PPAD does not require calcium for catalysis, it is only composed of a ~40 kDa catalytic domain, and it can convert free L-arginine to free L-citrulline [61,148,149]. In addition, PPAD only modifies C-terminal arginine residues (i.e., an exodeiminase), as those generated after peptide cleavage by arginine gingipain (Rgp) [149], which is also secreted by *P. gingivalis*. Several PPAD variants have been identified through the analysis of *P. gingivalis* genomes and clinical isolates [150]. One of these variants (termed PPAD-T2) has two-fold higher catalytic activity compared with PPAD from the reference strain (PPAD-T1) [150]. Whether PPAD variants are relevant for *P. gingivalis* pathogenesis is unknown.

PPAD is detected in the outer membrane (OM) fractions of *P. gingivalis* and as a secreted enzyme, which is found in a soluble form and in association with outer membrane vesicles (OMVs) [120,151,152]. In most clinical isolates (termed type I isolates), extracellular PPAD is mainly found in secreted OMVs and to a minor extent in a soluble form. In contrast, a small subset of clinical isolates (termed type II isolates) showed minimal levels of PPAD in OMVs and most of the enzyme in the soluble form [152]. Type II isolates are associated with a lysine residue at position 373 in PPAD [152]. The clinical significance of *P. gingivalis* type I and type II subsets in the pathogenesis of PD and RA is unknown. Secreted PPAD is believed to be a major virulence factor of *P. gingivalis* due to its capacity to generate ammonia during deimination of arginine to citrulline. Ammonia may protect *P. gingivalis* during acidic cleansing in the mouth [153], and promote periodontal infection by inhibiting neutrophil function [154,155].

### 5.3. PPAD-Mediated Citrullination of Bacterial and Host Proteins

A potential association between PPAD and citrullination in RA was initially suggested as a hypothesis in 2004 [62]. The first experimental evidence, published in 2010, provided two major findings [156]. The first was that clinical isolates of *P. gingivalis* were highly enriched in citrullinated proteins, suggesting that Rgp and PPAD are actively cleaving and citrullinating a large number of substrates in *P. gingivalis*. The second was that after cleavage by Rgp, PPAD citrullinates C-terminal arginine residues in fibrinogen and α-enolase, two important targets of ACPAs in RA. Together, these data provided initial evidence to suggest that bacterial and host proteins citrullinated by PPAD might initiate the loss of tolerance to citrullinated autoantigens in RA [156].

Subsequent studies, however, did not confirm the abundant citrullination initially found in *P. gingivalis* [120,128]. Indeed, although PPAD may citrullinate some proteins in *P. gingivalis*, this feature appears to be exclusive to a few bacterial strains (it was only detectable in reference strain W83 and the clinical isolate MDS45), and potential citrullination seems to be limited to only six bacterial proteins [128]. Whether C-terminal citrullinated peptides derived from these bacterial proteins are specific targets of ACPAs is unknown. Interestingly, PPAD is stable at low pH, resistant to limited proteolysis, and retains significant activity after boiling [61]. Therefore, depending on the processing of *P. gingivalis* for protein analysis, it is possible that citrullination of bacterial proteins may occur in vitro following cell lysis [120]. As an artifact, this may explain the reports of abundant citrullination (including endocitrullination) in lysates from *P. gingivalis* [121,157], as well as the finding that some monoclonal ACPAs cross-react with citrullinated outer membrane antigens (OMAs) from *P. gingivalis* lysates [157].

Regarding autoantigen citrullination by *P. gingivalis* [156], it is important to highlight that neither Rgp nor PPAD have substrate specificity. The finding that fibrinogen and α-enolase are cleaved and citrullinated by these enzymes is, therefore, not surprising, as any other protein would be predicted to undergo the same processing when exposed to these enzymes. Since cleavage and citrullination by Rgp and PPAD is not specific for RA autoantigens, additional evidence is required to support a role of PPAD in the lack of tolerance to citrullinated autoantigens. For example, this could be achieved by demonstrating that C-terminal citrullination mediated by Rgp and PPAD enhances immunoreactivity to autoantigens in RA.

### 5.4. Self-Endocitrullination of Pro-PPAD

PPAD is expressed as a pro-enzyme (thereafter pro-PPAD) that contains four domains: an N-terminal pro-peptide, a catalytic domain, an immunoglobulin-superfamily (IgSF) domain and a C-terminal domain (CTD) [148,149]. The mature enzyme only contains the catalytic and IgSF domains [148]. N-terminal processing appears to maintain enzyme activity and stability [120,158], while C-terminal cleavage is required for cell surface translocation of PPAD [159,160]. During the production and immunogenic analysis of recombinant pro-PPAD, it was noticed that the pro-enzyme undergoes self-endocitrullination when expressed in *E. coli,* and that this modified protein was preferentially recognized by RA sera when compared with the native pro-enzyme [121]. Using 13 cyclic PPAD peptides (termed CPP1-CPP13, which should not be confused with cyclic citrullinated peptides or CCPs) encompassing 18 potential citrullination sites in pro-PPAD, the study further identified immunodominant endocitrullinated peptides recognized by RA antibodies [121]. Although these results were surprising, because PPAD has no endocitrullination activity, it was assumed that these findings were analogous to PPAD from *P. gingivalis*. Thus, the data were interpreted to suggest that loss of tolerance to citrullinated proteins in RA may originate from an antimicrobial immune response directed against citrullinated PPAD [121].

Nevertheless, further studies demonstrated that self-endocitrullination of pro-PPAD was an artifact that resulted from the lack of N-terminal processing of the enzyme expressed in *E. coli* [120]. The absence of endocitrullination in PPAD has been additionally confirmed in structural studies using processed PPAD expressed in *E. coli* or generated in *P. gingivalis* [148,149]. In accordance with these findings, it was also demonstrated that although patients with RA have antibodies to PPAD, these antibodies have no preferential reactivity to the citrullinated pro-enzyme made in *E. coli* [120]. Together, these data demonstrated that self-endocitrullination of pro-PPAD in *E. coli* and the serum reactivity to this modified protein are unlikely to reflect the function and immunogenicity of PPAD from *P. gingivalis* [14,120,161].

Although some RA sera appear to react with cyclic citrullinated peptides derived from pro-PPAD (i.e., CPPs) [121,162], it is unclear whether these peptides are recognized by antibodies to native PPAD or if similar to commercial CCPs, CPPs may function as non-specific targets to detect ACPAs [161]. In this regard, although it was justified that CPPs were generated without knowledge that endocitrullination of pro-PPAD in *E. coli* was an artifact, there is no scientific support to continue their use in the study of RA. Nevertheless, antibodies to CPPs are still being used as biomarkers to link PPAD self-endocitrullination and RA pathogenesis. In particular, serum reactivity against the peptide termed CPP3 has been used to define clinical associations between *P. gingivalis* and RA [122,123,163,164], and to demonstrate that *P. gingivalis* can induce antibodies to citrullinated PPAD in rats [143]. Since the significance of CPPs is unclear, the findings generated with these peptides should be taken with caution. More importantly, serum reactivity to CPPs should not be considered as a maker of citrullination induced by PPAD.

### 5.5. PPAD in Experimental Models of Arthritis

To establish a direct role of PPAD in the production of ACPAs in RA, several experimental models of arthritis have been used to demonstrate that *P. gingivalis* induces or exacerbates arthritis *via* PPAD-mediated autoantigen citrullination and ACPA production. However, while several studies appear to confirm this hypothesis [131,138,144], these studies also highlight important misunderstandings in the study of protein citrullination and ACPAs in RA. For example, in a model of *P. gingivalis* and CIA, citrullinated proteins were mistakenly quantified by a colorimetric method, which measures both free citrulline and peptidylcitrulline [131]. In addition, unconventional methods have been used to detect or define ACPAs. This include the quantification of IgG levels against ACPA (i.e., anti-ACPA antibodies) rather than antibodies to the citrullinated substrates themselves [138]. More recently, one study showed that *P. gingivalis* drives ACPAs (detected by the anti-CCP2 assay) and erosive arthritis in Lewis rats, providing the first evidence that this microbial agent may be arthritogenic [144]. However, it also demonstrated that the antibody response was not specific to the citrullinated forms of antigens [144]. 

The failure to demonstrate a role of PPAD in autoantigen citrullination and ACPA production in mice and rats is not surprising. Although these animal models express synovial citrullinated proteins and recapitulate several features of the human disease [140,165], they do not produce antibodies specific for citrullinated proteins as observed in RA [165,166,167,168,169]. Moreover, they have a high amount of false-positive reactivity toward citrullinated peptides [166], which is frequently misinterpreted as ACPA positivity when non-citrullinated peptide controls are not included to confirm specificity [131,139,140]. Considering these important shortcomings, even if PPAD plays a critical role in the induction of ACPAs in RA, this hypothesis will be hard to demonstrate using current models of immune-mediated arthritis. Indeed, the data from some animal models of arthritis strongly suggest that *P. gingivalis* may exacerbate or induce disease by mechanisms independent of ACPA production, which may involve the activity of PPAD as a virulence factor targeting either free L-citrulline or protein substrates [131]. Thus, although epidemiological and experimental data may suggest a potential role of *P. gingivalis* in RA pathogenesis, there is no experimental evidence to support that this process is mediated by the induction of ACPAs against host or bacterial proteins citrullinated by PPAD. 

## 6. *Aa* in RA Pathogenesis

### 6.1. Infection Due to Aa

*Aa*, initially named *Bacterium actinomycetem comitans*, is a Gram-negative oral pathobiont first described in 1912 as a co-isolate from actinomycosis lesions [170]. It was reclassified as *Actinobacillus actinomycetemcomitans* and *Haemophilus actinomycetemcomitans* [171], and finally transferred in 2006 to a new genus, the *Aggregatibacter* [172]. For many years, it was thought that *Aa* was unable to cause infection, except in conjunction with *Actinomyces* [173,174]. This idea was initially supported by the finding that injection of a pure culture of *Aa* in the skin of a normal individual produced no infection [174]. In the 1960’s, however, several cases of *Aa* infection not associated with actinomycosis were reported, which were mainly associated with endocarditis [175,176,177]. *Aa* is now considered part of the HACEK (*Haemophilus*, *Aggregatibacter* (previously *Actinobacillus*), *Cardiobacterium*, *Eikenella*, *Kingella*) group of bacteria that are an unusual cause of infective endocarditis [178]. In addition, it is recognized as a cause of serious infections that can affect almost any organ, although these are very uncommon [179,180]. 

Although *Aa* is a rare cause of infection, this bacterium has attracted special attention since the 1980s because of its strong association with severe forms of periodontitis [181,182,183,184,185], both localized aggressive periodontitis (LAP) and chronic periodontitis [181,182,186,187,188,189,190,191,192,193,194,195]. *Aa* expresses several virulence factors [196]. Among these, the production of leukotoxin A (LtxA) is considered the major pathogenic component in the progression of aggressive periodontitis [195]. LtxA is a member of the repeats-in-toxin (RTX) family of pore-forming proteins [197]. Lymphocyte function-associated antigen (LFA)-1 (CD11a/CD18), Mac-1 (CD11b/CD18) and α_X_β_2_ (CD11c/CD18) act as receptors for LtxA (CD18 harbors the major binding site for LtxA), which accounts for the selective killing of human leukocytes [198]. By secreting LtxA, *Aa* induces cytolysis of target cells, thereby disabling host immune defenses and permitting escape from immune surveillance. *Aa* isolates exhibit variable virulence potential [195]. Strains of the serotype b JP2 genotype, characterized by a 530-bp deletion in the promoter region of the leukotoxin operon, has been shown to be highly virulent [199,200]. This genotype, which expresses 10- to 20-fold-higher levels of LtxA due to deletion of a transcriptional terminator [201], is associated with LAP primarily in subjects of African descent [202]. A more detailed analysis of the pathogenic mechanisms associated with the induction of periodontitis by *Aa* have been reviewed recently. [203] 

### 6.2. Aa-Induced Hypercitrullination and the Production Citrullinated RA Autoantigens

A role of *Aa* in the pathogenesis of RA was suggested in 2016, during the search for periodontal pathogens that may explain the presence of hypercitrullination in periodontitis [92]. The study of gingival tissue from patients with PD using anti-citrulline antibodies provided initial evidence that protein citrullination is increased in PD [204,205], supporting the idea that periodontitis is a potential source of citrullinated autoantigens. In addition, the data suggested the possibility that this process was linked to the activity of PPAD [205]. However, although further analysis by mass spectrometry (MS) of GCF confirmed that protein citrullination is increased in PD, this approach also provided two novel findings [92]. First, GCF from patients with PD is highly enriched in citrullinated proteins, mimicking patterns of cellular hypercitrullination found in the rheumatoid joint [92]. Second, peptide spectra of citrullinated proteins in periodontitis were consistent with endocitrullination [92], suggesting that an abnormal activation of host PADs, rather than PPAD, may be responsible for this process. Importantly, the presence of endocitrullination in periodontitis has been further confirmed by MS in GCF and periodontal tissue from patients with PD, both with and without RA [93].

Among different microbial species associated with severe periodontitis, in vitro studies identified *Aa* as the only pathogen that could reproduce the citrullinome found in periodontitis and RA, including the production of well-known targets of ACPAs [92]. In contrast to *P. gingivalis*, however, *Aa* does not encode a PAD-like enzyme. Instead, *Aa* drives citrullination by hyperactivating host PADs through the activity of LtxA. This toxin targets neutrophils, which are enriched in PAD enzymes and constitute the major immune cells in PD and the RA joint [206,207,208]. Since PADs are calcium dependent, the prominent calcium influx generated by the lytic effect of LtxA hyperactivates these enzymes, driving global hypercitrullination of a wide range of cellular proteins [92]. During this process, cell membrane integrity is eventually lost and membrane rupture occurs with release of citrullinated contents into the extracellular space [92]. Interestingly, this process is similar to the induction of hypercitrullination by host immune-effector pathways (i.e., perforin and complement) in the RA joint [87], suggesting that membranolytic damage by pore-forming proteins may be a unifying mechanism in the production of citrullinated autoantigens [209]. The form of cell death induced by these pore-forming mechanisms has recently been named leukotoxic hypercitrullination (LTH) [209], to distinguish it from other forms of neutrophil death that do not induce hypercitrullination.

### 6.3. Aa Exposure and RA Pathogenesis

Similar to *P. gingivalis*, epidemiological studies to identify an association between *Aa* and RA have been done by detecting antibodies in serum. In particular, IgG antibodies to LtxA (used as markers of *Aa* infection) have been measured in two large cohorts of patients with RA [92,210]. In one study that included patients with established RA, anti-LtxA antibodies were significantly associated with RA when compared to healthy individuals without PD [92]. Similarly, anti-LtxA antibodies were associated with chronic periodontitis in patients without RA, and the strongest association was observed in individuals with severe periodontitis [92]. Antibodies to LtxA may, therefore, identify a subgroup of RA patients with moderate to severe PD. Interestingly, this study also found that the association between *HLA-DRB1* susceptibility alleles (known as shared epitope alleles) and RA autoantibodies was restricted to patients who had evidence of *Aa* exposure. This suggested that in susceptible individuals, LtxA-induced hypercitrullination may play a role in ACPA production [92]. In support of this hypothesis, ACPA production and RA-like symptoms were recently reported in a patient with *Aa* endocarditis who had strong a genetic susceptibility to RA conferred by three HLA alleles linked to ACPA-positive RA [211]. 

In a different study using a large cohort of patients with early RA, anti-LtxA antibodies were also significantly enriched in RA compared to healthy controls and patients with other inflammatory arthritides (in both groups PD status was unknown) [210], supporting the high prevalence of *Aa* exposure in RA, both in early and established disease. However, this study found that the interaction between *HLA-DRB1*-SE and anti-CCP positivity was not exclusive to anti-LtxA-positive patients with RA [210]. Major differences among these cohorts may explain this discrepancy [212].

Analogous to *P. gingivalis*, metagenomic sequencing and PCR analysis in subgingival plaque have not found a significant association between the presence of *Aa* and RA [104,105,108,110]. In one study, however, analysis of GCF using commercial DNA probes (micro-Ident) identified *Aa* as the only periodontal pathogen showing significant differences between RA and non-RA controls (both with and without PD) [107]. Thus, similar to *P. gingivalis*, positive associations with *Aa* likely indicate PD severity irrespective of RA status.

Few studies have explored the potential role of *Aa* in experimental arthritis [142,213]. However, defining the causal effect of *Aa* in animal models of arthritis has the same limitations as those described for the study of *P. gingivalis*. Moreover, LtxA is known to have activity against leukocytes in primates, with no toxicity on rodent cells [214]. Therefore, different to the human model, any effect that *Aa* may have in the induction of experimental arthritis in rodents is unlikely to be driven by LtxA-induced hypercitrullination and the production of ACPAs.

## 7. *P. gingivalis* and *Aa* in the Mechanistic Model of RA: Causal Agents, Risk Factors, Disease Modulators, or Research Distractors

Assuming that periodontitis has a mechanistic role in RA, *P. gingivalis* and *Aa* may influence disease pathogenesis at different phases of RA development. The production of autoantibodies is an early and asymptomatic event that precedes the clinical onset of RA by several years [215,216]. The presence of a chronic and amplifying immune response against bacterial-induced RA autoantigens, in particular citrullinated antigens is, therefore, one of the most attractive hypothesis to explain disease initiation and potentially a causal association. In this context, this model explains why there is so much interest in establishing a causal relationship between *P. gingivalis* and the production of ACPAs in either experimental arthritis or RA. Although the current evidence neither demonstrate nor exclude the possibility that PPAD generates citrullinated autoantigens, other possibilities may explain a potential role of *P. gingivalis* in RA pathogenesis. Interestingly, protein citrullination is a process that is increased during inflammation [217]. However, while transient inflammation may not be sufficient to initiate an immune response to citrullinated proteins, periodontitis is a chronic non-fatal illnesses that can start during adolescence and progress over decades [218]. During this time, it is possible that some individuals at risk may develop autoantibodies to citrullinated proteins. *P. gingivalis,* a keystone microorganism in periodontitis, is certainly a strong candidate for promoting chronic inflammation [100], which may indirectly drive the production of autoantibodies in susceptible individuals.

In contrast to *P. gingivalis*, a potential role of *Aa* in the lack of tolerance to RA-associated autoantigens involves the induction of lytic hypercitrullination [92]. In this regard, different factors such as *Aa* virulence (i.e., LtxA production) and bacterial load may define the risk of developing autoantibodies in the context of *Aa* infection in susceptible individuals [92,211]. Importantly, as other bacterial species also produce pore-forming toxins [219], it is possible that other pathogens may induce hypercitrullination and ACPAs at other mucosal sites, including the lung and gut [1,209]. LTH as a mechanism of autoantigen production by different pathogens may explain why every patient with seropositive RA does not have exposure to *Aa* and/or PD. 

Interestingly, patients with RA show abnormalities in central and peripheral B cell tolerance checkpoints [220,221,222], likely due to an intrinsic genetic predisposition [223,224]. This results in the accumulation of autoreactive and polyreactive B cells that can recognize immunoglobulins and citrullinated peptides [220,221]. Depending on genetic and environmental risk factors, autoreactive or polyreactive naïve B cells may have a selective advantage to develop into high affinity autoreactive memory B cells through somatic mutations and affinity maturation. In this scenario, autoantigens generated by arthritogenic bacteria may promote the expansion and maturation of already existing autoreactive cells, rather than initiating the loss of tolerance to host antigens. A prerequisite of having autoreactive/polyreactive B cells to induce pathogenic autoantibodies by *P. gingivalis* or *Aa* may explain why only a limited number of individuals with PD may develop RA.

Although the hypothesis that periodontal pathogens (e.g., *P. gingivalis* or *Aa*) are causal agents of RA is an attractive idea, it is also possible that these bacteria may only be relevant as risk factors for the development of ACPA-positive RA. Alternatively, once RA has been established, persistent low-grade inflammation associated with PD may modulate RA progression and severity. Lastly, despite the enormous enthusiasm of demonstrating that periodontitis is relevant for RA pathogenesis, it cannot yet be excluded that patients with RA may have a higher risk of developing PD (even during the pre-clinical phase of RA), but neither periodontitis, *P. gingivalis*, nor *Aa* may play a pathogenic role in the disease.

## 8. Therapeutic Implications

The possibility of identifying a factor that triggers a multifactorial disease, such as RA, has critical implications for treatment and preventive interventions. The success of any therapy, however, may depend on targeting such a factor at the correct time during the evolution of the disease [225]. In this regard, although a systematic review and meta-analysis suggested that periodontal treatment may decrease some markers of disease activity in RA [226], these findings remain controversial [227]. The lack of an efficient response to PD treatment on RA disease activity may suggest that periodontitis has no pathogenic role in RA, or that it is most important for disease initiation, during the pre-clinical phase, and that treatment during established disease is no longer effective. If a periodontal pathogen is relevant for the initial breech of tolerance to arthritogenic autoantigens, aiming to treat asymptomatic autoantibody positive (e.g., anti-CCP, RF, or any other autoantibody) individuals may still be too late to stop the progression to symptomatic RA. In this scenario, preventive therapies (e.g., oral hygiene and potentially vaccines against periodontal pathogens) should be initiated in at-risk individuals prior to the development of these serologies. In contrast, if periodontitis is important for the amplification of the autoimmune response in preclinical RA, there is a window of opportunity to target seropositive asymptomatic individuals. Indeed, to date, this is the most feasible hypothesis that could be therapeutically addressed. However, if periodontal pathogens only play a role in the induction of specific subtypes of autoantibodies (e.g., ACPAs), eradication of these agents during the pre-clinical phase of the disease may only decrease the risk of developing certain autoantibodies (and potentially further RA severity), but may not change the course of the disease, including the production of antibodies to other autoantigens. Interestingly, recent evidence suggest that periodontitis is a potential source of proteins modified by malondialdehyde-acetaldehyde adducts and carbamylation [228], which are also common targets of autoantibodies in RA. [229] Thus, independent of the periodontal pathogen, preventing, or targeting periodontitis may be the best exploratory option to prevent RA.

## 9. Conclusions 

Although some epidemiological and mechanistic data supports a role for *P. gingivalis* and *Aa* in the pathogenic model of RA, it is important to underscore that these oral pathobionts do not fulfill the Henle-Koch’s postulates of causation [230,231,232]. These microbial agents neither occur in every case of the disease, nor are specific for patients with RA. In the context of inducing ACPA production in vivo to satisfy the third postulate, it will require the use of animal models that can truly reproduce the autoantibody response observed in RA. Considering these shortcomings, demonstrating or discarding a role of *P. gingivalis* and *Aa* in the pathogenesis of RA may require a different view of how to define disease causality by these microbial agents [212].

In the case of *P. gingivalis*, it is imperative to recognize that its potential role in RA involves elucidation of at least three different questions. The first is whether *P. gingivalis* is relevant for RA pathogenesis, either by initiating or exacerbating the disease. The second is whether *P. gingivalis* is important for RA because of the production of PPAD or because it plays a critical role in the induction of periodontitis by inciting inflammation. Indeed, it appears that ligation-induced periodontitis (without additional infection with *P. gingivalis*) is sufficient to exacerbate CIA in Wistar rats [233]. The third is whether PPAD drives RA due to the production of citrullinated antigens and the induction of ACPAs in the host or because this enzyme is a virulence factor that facilitates bacterial growth and the establishment of PD. Although these questions have been somewhat addressed, definitive conclusions cannot be reached from the available data. Defining the role of PPAD in the production of citrullinated antigens and whether C-terminal deimination offers an advantage for peptide immunogenicity remains a high priority. Importantly, other potential arthritogenic mechanisms induced by *P. gingivalis*, besides autoantigen citrullination, should be kept in consideration. 

In the case of *Aa*, this model offers an advantage over other periodontal bacteria because of its capacity to induce neutrophil hypercitrullination and the release of citrullinated autoantigens through osmotic lysis [92]. However, the number of studies linking *Aa* with RA are too limited to truly determine its pathogenic significance for the disease. Similar to *P. gingivalis*, rigorous studies are necessary to support or refute the role of *Aa* in the etiology of RA. Together, these oral pathobionts pose an opportunity to understand whether bacterial-associated citrullination is a mechanism involved in RA pathogenesis. This has important implications for the implementation of preventive interventions in RA.

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
