# Peer review of "Rheumatoid Arthritis-Associated Mechanisms of Porphyromonas gingivalis and Aggregatibacter actinomycetemcomitans"

_jcm, 2019, doi:10.3390/jcm8091309_

Round 1

Reviewer 1 Report

This review is a comprehensive overview of the association of two bacteria which cause PD and RA, therefore, providing a balanced argument for why these mechanisms instrumented by these bacteria may explain the association between PD and RA. Although a variety of different studies and hypotheses were explored in this review, the authors have rightly concluded that there is still not enough evidence to suggest these microbes play a convincing role in RA pathogenesis. The context to the main subject of the review is presented with detail. Some minor comments below:

Page 4, line 90, references required Page 6, line 120-123, add effect sizes and p-values reported in these studies Page 9, 196, typo- should be "recommended", not recommend Pae 17, line 419 typo- should be "In support of..." not "In support to..."

Author Response

Reviewer #1

Comment 1: Page 4, line 90, references required

Response: The reference has been added (ref #24).

Comment 2: Page 6, line 120-123, add effect sizes and p-values reported in these studies

Response: The values have been added (lines 154 and 155).

Comment 3: Page 9, 196, typo- should be "recommended", not recommend

Response: The typo was corrected.

Comment 4: Page 17, line 419 typo- should be "In support of..." not "In support to..."

Response: The typo was corrected.

Reviewer 2 Report

I read with great interest the paper from E Gomez-Banuelos et al. entitle “Rheumatoid arthritis-associated mechanisms of Porphyromonas gingicalis and Aggregatiber actinomycetemcomitans

In this paper, the authors reviewed extensively association between RA and periodontal disease and role of two dental germs. This is a very nice work.

However, I have few major comments:

In the Summary, authors finished this part by “Here, we will discuss the clinical and mechanistic evidence surrounding the role of these periodontal bacteria in RA pathogenesis, which highlights a key area for the treatment and 30 preventive interventions in RA”. This is right that the first part of the sentence was nicely performed. However, the last part was not discussed in the manuscript and should be discussed at the end as “perspective”. I would suggest to read an editorial on this purpose (doi: 10.1001/jamanetworkopen.2019.5358). Authors forget to cite this negative paper (Zhang X et al. Nature Medicine 2015;21: 895–905), where Porphyromonas gingicalis was not found in RA salivary and discussed pitfall explaining why Porphyromonas gingicalis was not observed. At least one paper (doi: 10.1186/s12967-018-1588-2) suggested that gingival citrullination was due to chronic inflammation independently to Porphyromonas gingicalis or Aggregatiber actinomycetemcomitans. Evidence of implication of Aggregatiber actinomycetemcomitans compared to Porphyromonas gingicalis is still debated and needs to be more extendedly discussed. In the Chapter 3, I suggest to discuss the same pathogenesis between both diseases (shared epitope, smoking, Th17,….). Since both diseases are linked, a treatment of one disease could impact the status of the second one. I suggest to discuss this paper (doi: 10.3390/jcm8050751).

Author Response

Reviewer #2

Comment 1: In the Summary, authors finished this part by “Here, we will discuss the clinical and mechanistic evidence surrounding the role of these periodontal bacteria in RA pathogenesis, which highlights a key area for the treatment and 30 preventive interventions in RA”. This is right that the first part of the sentence was nicely performed. However, the last part was not discussed in the manuscript and should be discussed at the end as “perspective”. I would suggest to read an editorial on this purpose (doi: 10.1001/jamanetworkopen.2019.5358).

Response: A new section (#8) has been included to discuss therapeutic implications. The paper doi: 10.1001/jamanetworkopen.2019.5358 has been cited in the manuscript (ref # 224).

Comment 2: Authors forget to cite this negative paper (Zhang X et al. Nature Medicine 2015;21: 895–905), where Porphyromonas gingicalis was not found in RA salivary and discussed pitfall explaining why Porphyromonas gingicalis was not observed.

Response: The paper by Zhang X et al has been included (reference 109) and it is discussed on pages 9 and 10. Because of the inclusion of this paper, we needed to discuss additional publications that have also showed controversial associations between RA and both for P. gingivalis and Aa (pages 9-11 and 21, respectively). Pitfalls are also discussed.

Comment 3: At least one paper (doi: 10.1186/s12967-018-1588-2) suggested that gingival citrullination was due to chronic inflammation independently to Porphyromonas gingicalis or Aggregatiber actinomycetemcomitans.

Response: The reviewer is correct. However, we found some potential caveats to discuss this paper. In this manuscript, Engström et al detected the presence of P. gingivalis and Aa-LtxA in gingival tissue by staining using antibodies. Besides that the authors showed no evidence of how these antibodies were validated in tissues (e.g. by showing negative and positive controls), we were unable to find references demonstrating that tissue staining is an efficient method to confirm the presence of these pathogens in gingival tissue from patients with periodontitis compared with no-PD. Indeed, in the study by Engström et al, neither P. gingivalis nor Aa-LtxA staining showed differences in PD compared with no-PD, which may suggest the possibility that the staining is not specific. Because of this potential caveat, it is difficult to reach a conclusion regarding the association between citrullination and the presence of periodontal pathogens. We completely agree that periodontal inflammation in likely enough to drive citrullination independently of P. gingivalis or Aa, as was discussed in our review. However, we would prefer not to include the manuscript by Engström et al to support this idea.

Comment 4: Evidence of implication of Aggregatiber actinomycetemcomitans compared to Porphyromonas gingicalis is still debated and needs to be more extendedly discussed.

Response: The reviewer is correct. However, we believe that the implications of each pathogen should be presented independently of each other. We consider that the debate regarding Aa or P. gingivalis in RA should not be focused on fighting who is correct or incorrect. Instead, the debate should be centered on the quality of the research surrounding each pathogen. P. gingivalis provides an excellent model to explain much of the pathology in PD that can be translated to RA. The problem with P. gingivalis is that association studies are inconsistent and studies trying to demonstrate that this bacteria is responsible of driving ACPAs (via citrullination of autoantigens by PPAD) have important limitations. This is not the fault of Aa. In the case of Aa, it provides an excellent model for autoantigen citrullination. However, the limitation has been to confirm that only individuals with genetic risk (i.e. HLA-DR-SE) are susceptible to develop ACPAs after Aa infection. The insufficient number of studies about Aa and RA is also an important limitation, but this is not the fault of P. gingivalis. In the review, we tried to discuss Aa and P. gingivalis separated to avoid the idea that they are competitors. We tried to provide an unbiased analysis of the published data including the pros and cons for each pathogen. Our intention is to spark the interest of the readers to question the work that has been done for both pathogens. We believe that by comparing Aa vs. P. gingivalis, this focus may be lost. In addition, we have the caveat that our work is about Aa, which may add an additional bias to the reader if we compare both pathogens.

Comment 5: In the Chapter 3, I suggest to discuss the same pathogenesis between both diseases (shared epitope, smoking, Th17,….).

Response: A discussion about pathogenic similarities between PD and RA has been included in section 3 (lines 134-142).

Comment 6: Since both diseases are linked, a treatment of one disease could impact the status of the second one. I suggest to discuss this paper (doi: 10.3390/jcm8050751).

Response: We agree that targeting common mechanisms in PD and RA should provide mutual benefit to both diseases by decreasing downstream damage associated with the pathway that is being targeted. For example, if an important inflammatory cytokine is therapeutically blocked, the downstream benefits may include decrease in inflammation, bone damage, downstream cytokines, metalloproteinases, autoantibodies, etc. However, the effect of P. gingivalis is an upstream event that drives the induction of inflammation and PD. Thus, by blocking inflammatory pathways downstream of P. gingivalis (e.g. TNF), why should be expected that the bacterial load is going to decrease? Indeed, the induction of PD is a complication of an uncontrolled inflammatory response that was initially driven to stop bacterial proliferation. Thus, by blocking cytokines that are trying to decrease the load of P. gingivalis, it is expected that this may indirectly result in more bacterial proliferation.

In the manuscript by Rinaudo-Gaujous et al (doi: 10.3390/jcm8050751), they addressed the effect of infliximab on markers of PD severity. Among these markers, they studied anti-P. gingivalis antibodies, which were used as indirect markers of bacterial load. It appears that the authors were expecting that blocking TNFα may have some antibacterial effect on P. gingivalis. A major conclusion of this paper is that in RA patients with PD, Infliximab did not decrease P. gingivalis load based on anti-P. gingivalis antibody titers. A second conclusion is provided in the title, “Infliximab Induced a Dissociated Response of Severe Periodontal Biomarkers in Rheumatoid Arthritis Patients”, which is not discussed in the manuscript. Indeed, the data showed that Infliximab significantly increases anti-P. gingivalis antibodies at 6 months after treatment. Thus, as expected, blocking TNFα appears to facilitate P. gingivalis proliferation. This finding has nothing to do with “Dissociated Response of Severe Periodontal Biomarkers in RA”. It is just the consequence of targeting a cytokine that is trying to control the load of P. gingivalis. Because of the long rationale to discuss this paper, we would prefer not to include it in the review.